# Potential Role of Fenestrated Septa in Axonal Transport of Golgi Cisternae and Gap Junction Formation/Function

**DOI:** 10.3390/ijms24065385

**Published:** 2023-03-11

**Authors:** Camillo Peracchia

**Affiliations:** Department of Pharmacology and Physiology, School of Medicine and Dentistry, University Rochester, Rochester, NY 14642-8711, USA; camillo.peracchia@gmail.com

**Keywords:** axons, nerves, axonal transport, axoplasmic flow, kinesin, Golgi apparatus, crayfish, giant axons, gap junctions, innexins, connexins

## Abstract

Crayfish axons contain a system of parallel membranous cisternae spaced by ~2 μm and oriented perpendicularly to the axon’s long axis. Each cisterna is composed of two roughly parallel membranes, separated by a 150–400 Å wide space. The cisternae are interrupted by 500–600 Å pores, each occupied by a microtubule. Significantly, filaments, likely made of kinesin, often bridge the gap between the microtubule and the edge of the pore. Neighboring cisternae are linked by longitudinal membranous tubules. In small axons, the cisternae seem to be continuous across the axon, while in large axons they are intact only at the axon’s periphery. Due to the presence of pores, we have named these structures “Fenestrated Septa” (FS). Similar structures are also present in vertebrates, including mammals, proving that they are widely expressed in the animal kingdom. We propose that FS are components of the “anterograde transport” mechanism that moves cisternae of the Golgi apparatus (GA) toward the nerve ending by means of motor proteins, likely to be kinesins. In crayfish lateral giant axons, we believe that vesicles that bud off FS at the nerve ending contain gap junction hemichannels (innexons) for gap junction channel and hemichannel formation and function.

## 1. Introduction

Axoplasmic transport is a function that enables the intra-axonal flow of membranous organelles, as well as proteins, lipids, and other molecules (rev. in [1,2,3]). The flow is bidirectional: transport toward the nerve ending is called “anterograde transport”, while transport toward the neuronal cell body is known as “retrograde transport”. Loads of cytoplasmic components such as mitochondria, vesicles, membranous cisternae, soluble proteins, and lipids are transported at different rates [4]. Membranous components are transported at the relatively fast rate of 50–400 mm/day, while soluble proteins and lipids move at ~8 mm/day.

Two motor proteins: kinesin and dynein, participate in the axoplasmic transport mechanism. Kinesin is involved in anterograde axonal transport, while dynein participates in retrograde axonal transport (rev, in [5]); both proteins link intracellular material to microtubules or microfilaments. As a hypothesis, we propose that parallel cisternae reported in axons of crayfish and other animals [6] are cisternae of the Golgi apparatus (GA) that flow toward the nerve ending by the anterograde transport mechanism. Both kinesin [7,8] and dynein [9] are known to bind to GA’s cisternae, but in our case kinesin rather than dynein is most likely the motor protein involved in this transport mechanism. The human genome contains forty-five kinesin genes of which thirty-eight are present in brain tissue; of them, kinesins-1, -2, and -3 are involved in anterograde axonal transport (rev. in [2]).

Do axons contain elements of GA? While ultrastructural studies have not reported the existence of typical elements of GA [10,11], axons are known to contain molecules of GA that are involved in secretion and protein synthesis. Indeed, inhibition of GA’s activity affects axonal growth [10,11]. It is generally believed that only vesicles that bud off trans-Golgi and ER cisternae are transported in the axoplasm; however, regular arrays of membranous cisternae have occasionally been reported in axons of vertebrates and invertebrates (see in the following).

Our early ultrastructural data have reported that various axons of crayfish abdominal nerve roots and ventral nerve cord contain regularly spaced and interconnected membranous cisternae which display small pores that are penetrated by microtubules linked to the pore’s edge by small filaments [6], likely to be composed by kinesin molecules. We believe that these structures, which we have named “Fenestrated Septa” (FS), are components of the anterograde axonal transport of GA’s cisternae [6,12]. In contrast, most studies that have described similar structures proposed that they are cisternae of the smooth endoplasmic reticulum (SER) [13,14,15,16,17,18,19], involved in the sequestration and storage of calcium ions [20], calcium homeostasis [19,21], or mobilization of intra-axonal Ca^2+^ [19].

Two studies have vaguely suggested that the cisternae may be involved in the transport of material throughout the axoplasm, but potential mechanisms of the transport system have not been proposed [15,20]. One of them has proposed that FS may play different roles in specialized SER, such as Ca^2+^ sequestration and Adenosine Triphosphate (ATP} or monoamine oxidase transport [20]. The other, which only reported an intracellular membranous system, and did not observe pores crossed by microtubules, only proposed an unspecified mechanism for transporting unidentified components in the axoplasm [15].

Curiously, most of the studies that have reported the existence of parallel cisternae have not noticed the presence of pores (Fenestrae) crossed by microtubules. Only two studies have noticed similar pore-like structures. In one study, the authors mistakenly interpreted the area surrounding the microtubules as non-membranous “dense material” [22] rather than the face view of the fenestrated cisterna. In the other study [20], the author suggested, as we did, that FS are involved in the axonal transport mechanism, but his interpretation of the nature of FS differed from ours, as he proposed that they are SER cisternae rather than GA cisternae. Significantly, no studies, aside from ours, have noticed the presence of filaments (likely to be kinesin molecules) linking the central microtubule to the edge of the pore.

Aside from crayfish, arrays of parallel membranous cisternae in the axoplasm have also been reported in other invertebrates, such as in the ganglia of *Armadillium vulgare* (Crustacea-Isopoda) [13], sinus gland of *Gammarus oceanicus* [19], scorpion ganglia and *Lepidoptera* embryonic nerve fibers [14], giant fibers of the walking leg of lobster and crayfish [15], stomatogastric ganglion of *Homarus Americanus* [23], the peripheral retina and *Lamina ganglionaris* of the fly *Musca domestica* [16], the axons of the flesh fly *Boettcherisca peregrina* [17], and the cardiac ganglion of prawn (*Penaeus japonicum bates*) [18]. However, FS would not be as relevant to our knowledge of axonal structure and function if they had not also been described in vertebrates, such as mammalian myelinated axons [6], frog brain [20], goldfish Mauthner cells [21], axons of the *Ammocoete larvae* of lamprey (*Petromyzon marinus*) [24], neurosecretory processes of the hypothalamus of the fish *Gasterosteous aculeatus* L. [25], chick spinal cord [22], mouse olfactory bulb [26], rat neurosecretory neurons, and other hypothalamic and hypophysial cells [27].

In view of recent advances in the field of axonal transport, we feel that it is important to bring back to light our early structural data on FS and to propose the hypothesis that the FS are GA cisternae transported along the axon toward the nerve ending, and that they participate in gap junction channel and hemichannel formation and function. We feel that most relevant in our early study is the observation that filaments link microtubules to the edge of FS’s pores. The likelihood that these filaments are kinesin molecules should be tested experimentally as it might be most relevant to our understanding of axonal transport mechanisms.

## 2. Structure of Fenestrated Septa

### 2.1. Phase-Contrast Microscopy of Fenestrated Septa

From each abdominal ganglion run three pairs of roots, which are designated first, second, and third, from cranial to caudal direction. The first and second roots contain both motor and sensory fibers, while the third root contains almost completely motor fibers. In phase-contrast microscopy, we noticed that axons sectioned longitudinally or slightly obliquely display regularly spaced striations (Figure 1 and Figure 2), which in the electron microscope correspond to parallel FS that cross the axoplasm in planes perpendicular to the long axis of the fiber (Figure 3).

In some areas, FS seem to cross the axoplasm without interruptions while in others there are pronounced discontinuities at the center of the axon. Uninterrupted FS are seen only when peripheral areas of the axoplasm are cut, while interrupted ones are seen where the axoplasm is cut near the center of the fiber (Figure 1). FS repeat every ~2 μm. The central region of the axons, where FS are not seen, show thin wavy filaments (Figure 1), which in the electron microscope prove to be elongated mitochondria oriented parallel to the long axis of the axon. FS are also present in larger axons such as median and lateral giant axons.

### 2.2. Electron Microscopy of Fenestrated Septa

A comparison between phase-contrast and electron microscope images of FS is shown in Figure 2 and Figure 3, which show the axons longitudinally sectioned in peripheral areas (Figure 2, inset). Figure 3 confirms that FS are made of cross-sectioned membranes separated by a narrow gap. Frequently, the membranes join creating a series of discontinuities (pores), some of which are crossed by a microtubule (Figure 4, arrows). FS are often seen to extend in longitudinally membranous tubules (Figure 5, T) which link them to adjacent FS.

Cross-sectioned axons display the face view of FS (Figure 6). In these images, the discontinuities seen in cross-sections (Figure 4) appear as small pores (Figure 6, Figure 7 and Figure 8), each occupied by a cross-sectioned microtubule (~250 Å in diameter). The pores are 500–600 Å in diameter (Figure 7 and Figure 8). In the smallest pores, usually circular, the microtubules are concentric with the pore (Figure 7), and occasionally thin filaments ~100 Å long are seen linking the microtubules to the edge of the pores (Figure 8, white arrows). Most likely, the filaments are kinesin molecules [5,28,29,30,31,32], which indeed are ~100 Å long (Figure 8, inset). The heavy chain of the crayfish kinesin, a 120 kDa protein, was isolated in 1996 by Okada and coworkers and named CF-kinesin [30].

Evidence that the pores seen in cross-sectioned axons represent the face view of FS is provided by images of obliquely sectioned axons, where FS display an intermediate view. Curiously, in one study, the face view images of FS were wrongly interpreted as non-membranous “dense material” [22]. In Figure 9, two axons are shown in different planes of section. In the top axon, which is cut longitudinally (inset A), the FS are cross sectioned as in Figure 3, while in the bottom axon, which is obliquely sectioned (inset B), the septa are organized in parallel fashion, spaced farther apart because of the obliquity of the section; here the pores start displaying their face view images as in Figure 6, Figure 7 and Figure 8.

FS are also seen in small fibers (Figure 10 and Figure 11). The relationship between pores and microtubules is clearly seen in these fibers as well (Figure 10, inset). Figure. 11 shows another small axon displaying a larger area of FS.

FS repeat every ~2 μm, a spacing that is like the repeat of T-tubules in skeletal muscles. This suggested the possibility, albeit remote, that FS might be invaginations of the plasma membrane as muscle T-tubules. To test this possibility, experiments were performed with the extracellular tracer horseradish peroxidase. As expected, peroxidase was found in the gap between axons and their adjacent satellite cells, but never in the lumen of the septa, proving that FS are not invaginations of the plasma membrane. A schematic representation of a small axon containing FS is shown in Figure 12.

FS are a general feature of axon structure (see in the previous). Indeed, they have also been seen, although in a less organized fashion, in mouse sciatic and vagus nerves (Figure 13 and Figure 14). In these axons, the pores are occupied by either neurofilaments (Figure 13) or microtubules (Figure 14).

### 2.3. Potential Role of Fenestrated Septa in Gap Junction Formation/Function

As a hypothesis, it is proposed that FS contain gap junction hemichannels (connexons/innexons) that at the nerve ending are transferred to the plasma membrane via vesicles that bud off the terminal cisternae of the FS. Indeed, in this review I describe the structure and function of gap junctions that join electrically and metabolically neighboring segments of lateral giant axons (Figure 15), focusing on accessory structures that may link FS to gap junction formation and function.

The crayfish nerve cord contains four giant axons, two median and two lateral (Figure 15A). While the median giant axons extend uninterruptedly along the entire nerve cord, the lateral giant axons are made of a series of axonal segments, each 2–3 mm long, that join in each ganglion at structures named “septa” (Figure 15(A,Ba)). Each septum consists of a sheet of fibrillar connective tissue (~1 μm thick) that is coated on both sides by satellite cells, whose cytoplasmic processes overlap each other either in close contact or are separated by thin layers of connective tissue (Figure 15B). In certain regions of the septum, large discontinuities in the combined satellite-cell–connective layer allow the surface membranes of the two axons to come in close apposition, forming gap junctions. In most cases, the junctions form between one of the two axons and finger-like processes of the other axon which find their way through the septum (Figure 15B).

In cross-sections, the gap junctions display a beaded profile reflecting the presence of gap junction channels, each composed of two innexon hemichannels ~125 Å in size (Figure 15(Bb)). In freeze fractured replicas, the innexons appear as 60–100 Å particles (Figure 16A) and complementary pits (Figure 16B,C). Particles and pits display central dimples (Figure 16A) and complementary bumps (Figure 16B), which represent the location of the channels’ pores (Figure 16C). The innexons are composed of six radially arranged innexin monomers, as seen in isolated negatively-stained junctions (Figure 17A–E) and in freeze-fracture replicas (Figure 17F,G) [33], and confirmed by the multiple rotation photographic exposure method of Markham and coworkers [34] (Figure 17B–G).

An interesting feature of gap junctions between crayfish lateral giant axons is the presence of 500–800 Å vesicles that line both junctional surfaces (Figure 15(Bb) and Figure 18) [35]. Significantly, the vesicles’ membrane often displays electron-opaque particles similar in size to the junctional particles (Figure 18A). The particles occasionally appear to precisely interact with the cytoplasmic end of junctional particles, forming what appears to be small intracellular junctions (Figure 18(Aa)). Indeed, particles such as gap junction particles are also seen in freeze-fractured vesicles’ neighboring gap junctions (Figure 19 and Figure 20, arrows); the particles and the complementary pits (Figure 19 and Figure 20, arrows) are similar in size to junctional particles and often display a similar central dimple (Figure 20, double-headed arrow). Occasionally, neighboring vesicles appear to directly bind to each other via particle–particle interaction (Figure 18(Aa)), suggesting that they may be intercommunicating.

Based on these observations, it has been suggested that the vesicles might be part of a direct communication between the internal compartment of vesicles lying on one side of the junction and that of vesicles lying on the other side of the junction (Figure 18B), apparently forming “intracellular gap junctions” (Figure 18B and Figure 21). If this were the case, compounds that must be excluded from the cytosol could be transferred to membrane-bound compartments of the other cell via three junctions, two intracellular and one intercellular (Figure 18B and Figure 21). It is understood that this hypothesis is very speculative because there is no experimental evidence that connexon/innexons can interact by their cytoplasmic domains. In 2020, the author suggested possible interaction between domains of the cytoplasmic loop of Cx32, Cx43 and innexin-1 [36], but once again these potential interactions are entirely speculative.

Interestingly, crayfish giant axons have two types of gap junctions. While the junctions described in the previous, which connect axonal segments of the lateral giant axon at the septa, allow bidirectional electrical and metabolic communication, gap junctions linking giant axons to motor fibers are rectifying junctions, as they only allow diffusion of the electrical impulse from giant (median or lateral) to motor axons. Like the bidirectional gap junctions of lateral giant axons, rectifying gap junctions also contain innexons which bind to each other across the extracellular gap. However, rectifying junctions are structurally asymmetrical, as vesicles only coat the presynaptic side of the gap junction [37]. Based on our hypothesis that proposes a vesicle-to-vesicle transfer of molecules (Figure 18B), it could be that in rectifying gap junctions, the content of presynaptic vesicles is transferred directly to the cytosol of the motor axon.

Our hypothesis is that the particles seen in the vesicles (Figure 19 and Figure 20) are innexons embedded in the membrane of vesicles that have budded off FS (cisternae) of GA (Figure 21). This is likely to be the case not only because they are identical in size to gap junctional innexons and contain small central dimples as well (Figure 20), but also because it is well known that most of the gap junction proteins (connexins and innexins) oligomerize into hexamers (connexons and innexons) in the trans-Golgi network [38,39,40], although a few oligomerize in the ER [41,42,43]. It is believed that some of the innexon-loaded vesicles that have budded off FS (Golgi cisternae) fuse with the plasma membrane (Figure 21, green arrow) forming innexon hemichannels (Figure 21, blue arrow), some of which eventually form cell–cell channels (Figure 21, GJ), while other innexon-loaded vesicles interact with gap junction membranes (Figure 21, red arrow, and Figure 18Ab).

## 3. Conclusions

In view of recent advances in the understanding of axonal transport, we felt the need to bring back to light in this review some structural elements (fenestrated septa, FS) that we published over half a century ago, because our hypothesis suggesting that they are part of the anterograde axonal transport mechanism has not been investigated and needs to be tested experimentally. FS were described in detail in crayfish axons [6], but were also identified in various vertebrate axons as well [21,22,24,25,27], suggesting a role in nerves in general.

Each cisterna of the FS is made of two parallel membranes, separated by a small gap. The two membranes are interrupted by pores 500–600 Å in diameter, each occupied by a microtubule. Filaments, most likely made of kinesin [28,29,30], are seen bridging the gap between the microtubule and the edge of the pore. FS are likely to transport various components assembled in the Golgi apparatus. However, in this review, we limit their potential role to that in the formation of gap junction channels and hemichannels.

For further determining the structure and function of FS, future experimental work needs to test the hypothesis that FS are indeed components of the GA, and if so, whether they are part of the anterograde or retrograde axonal transport. One may want to determine by immunofluorescence microscopy and/or immunoelectron microscopy whether FS contain proteins typical of GA or SER cisternae. Furthermore, the same methods should be used to determine the nature of the filaments that join the central microtubule to the edge of the pore, as they are likely to be the most relevant structural element of the axonal transport mechanism.

## Figures and Tables

**Figure 1 ijms-24-05385-f001:**
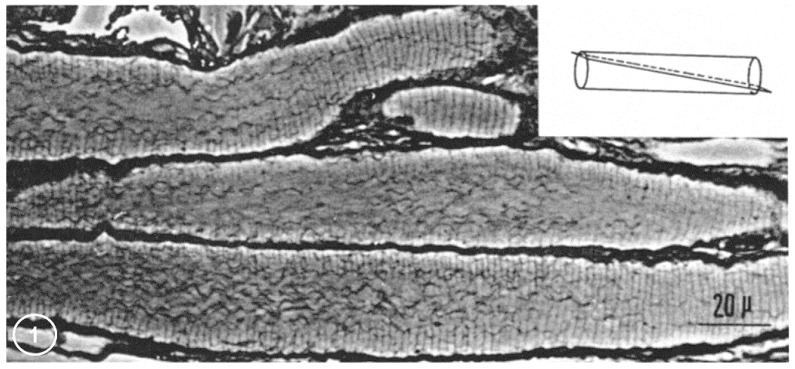
Phase-contrast micrograph of crayfish axons from the second root obliquely sectioned (see inset). The periphery of the axons displays parallel septa, spaced by ~2 μm. Central areas of the axons contain wavy filaments representing helically shaped mitochondria. Adapted from Ref. [6].

**Figure 2 ijms-24-05385-f002:**
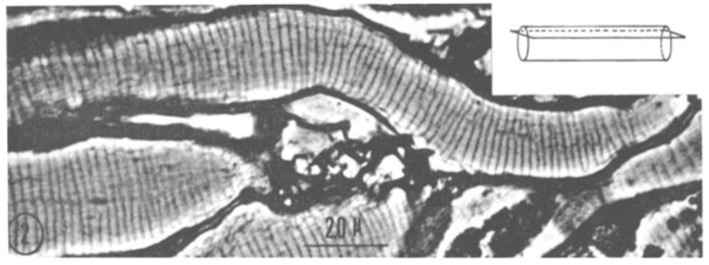
Phase-contrast micrograph of second-root’s axons longitudinally sectioned at the axon’s periphery (see inset), displaying cross-sectioned cisternae (Fenestrated Septa, FS). Adapted from Ref. [6].

**Figure 3 ijms-24-05385-f003:**
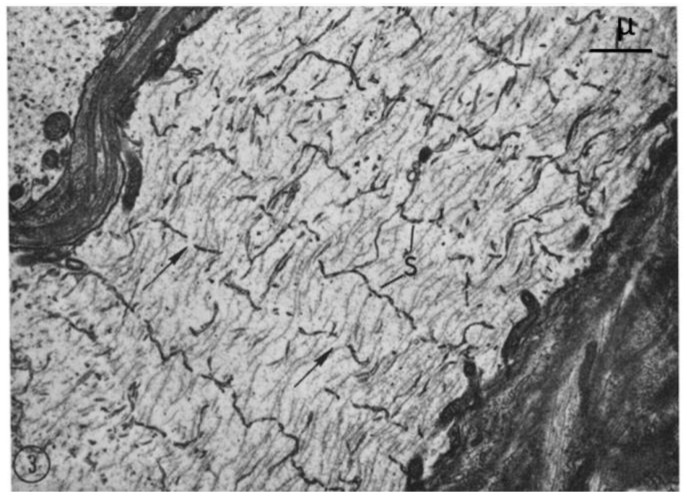
Electron micrograph of an axon longitudinally sectioned at the periphery showing cross-sectioned FS. Each septum (S) is made of two membranes separated by a 150–400 Å gap. The two membranes frequently join, creating 0.1–0.2 μm pores containing microtubules (arrows). Adapted from Ref. [6].

**Figure 4 ijms-24-05385-f004:**
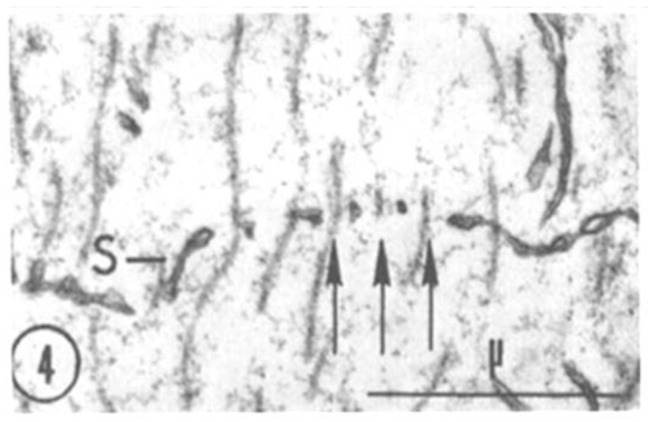
Detail of a cross-sectioned septum (S) displaying pores occupied by microtubules (arrows). Adapted from Ref. [6].

**Figure 5 ijms-24-05385-f005:**
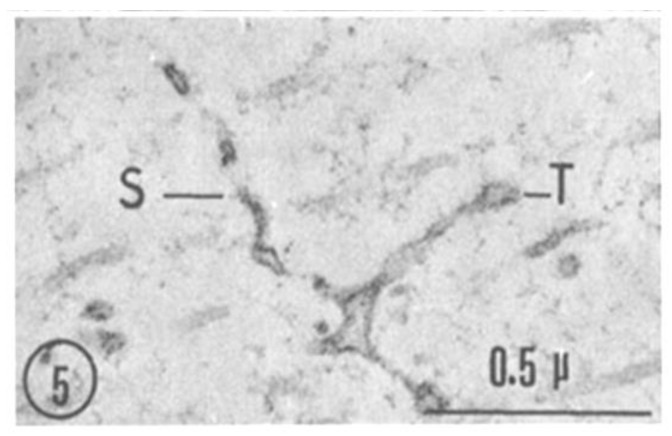
A septum extends into a membranous tubule (T) that connects it to a neighboring septum. Adapted from Ref. [6].

**Figure 6 ijms-24-05385-f006:**
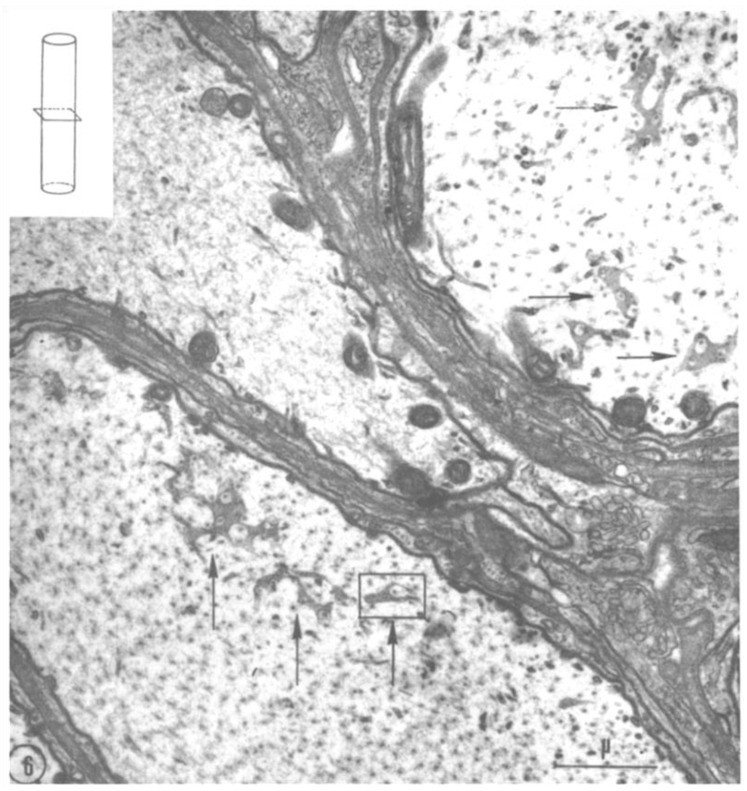
Electron micrographs of second root’s axons (~6 μm in diameter), transversely cut (seen inset). Face view images of fragments of fenestrated septa (arrows) display pores. Each pore contains a cross-sectioned microtubule. Adapted from Ref. [6].

**Figure 7 ijms-24-05385-f007:**
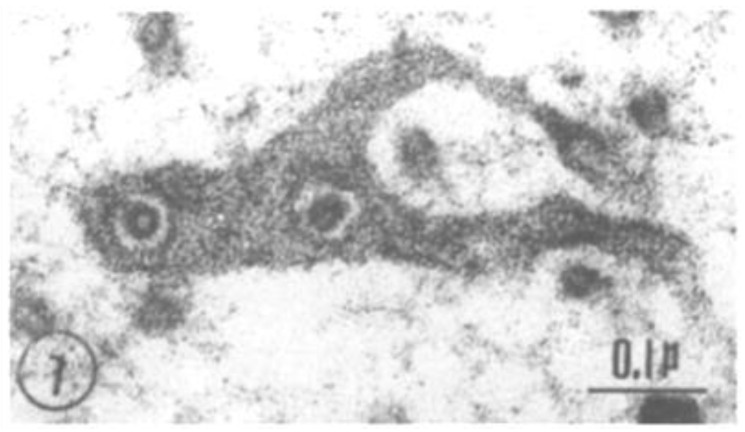
Detail of Figure 6 showing a pore, ~550 Å in diameter, containing a cross-sectioned microtubule; the gap between pore and microtubule is 80–100 Å wide. Adapted from Ref. [6].

**Figure 8 ijms-24-05385-f008:**
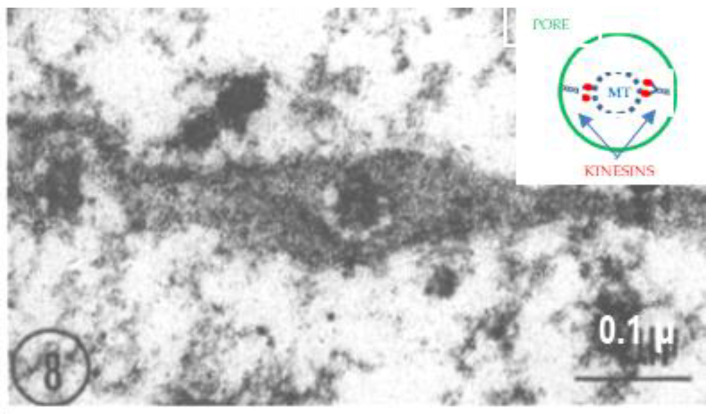
This pore, ~600 Å in diameter, displays thin filaments (white arrows) linking the microtubule (MT) to the edge of the pore (see inset). These filaments are likely to be kinesin molecules. Adapted from Ref. [6].

**Figure 9 ijms-24-05385-f009:**
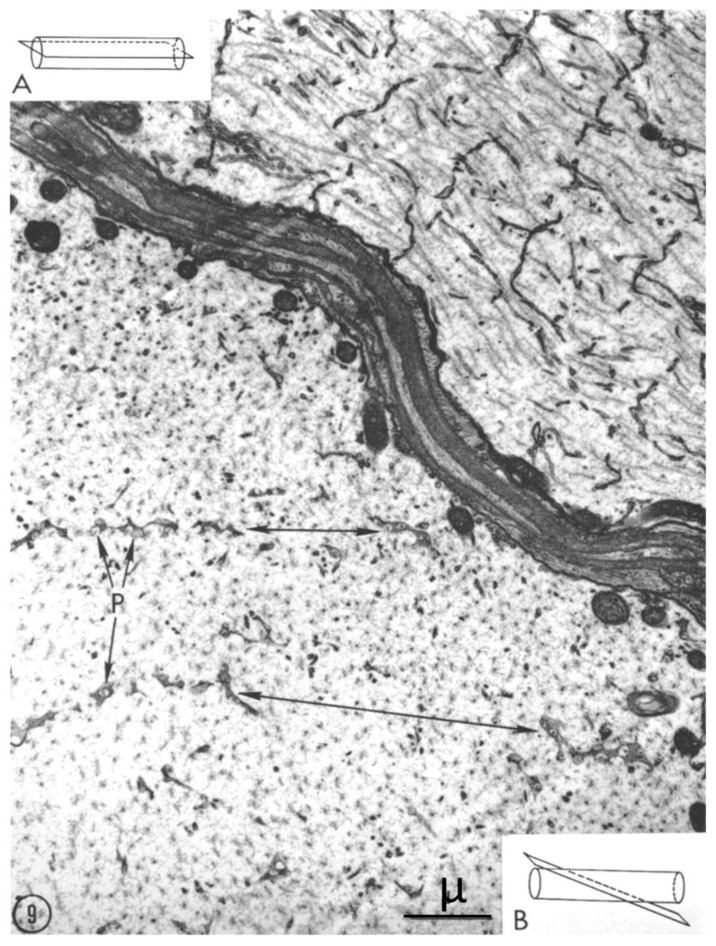
Axons of the second root (~10 μm in size). The top axon, longitudinally sectioned (inset (**A**)), shows a cross-section view of the septa. The bottom axon, obliquely sectioned (inset (**B**)), shows two parallel septa (double-pointed arrows) with partial face-views of the pores (P). The top and bottom axons are separated by an oblique structure composed of satellite cells and connective tissue. Adapted from Ref. [6].

**Figure 10 ijms-24-05385-f010:**
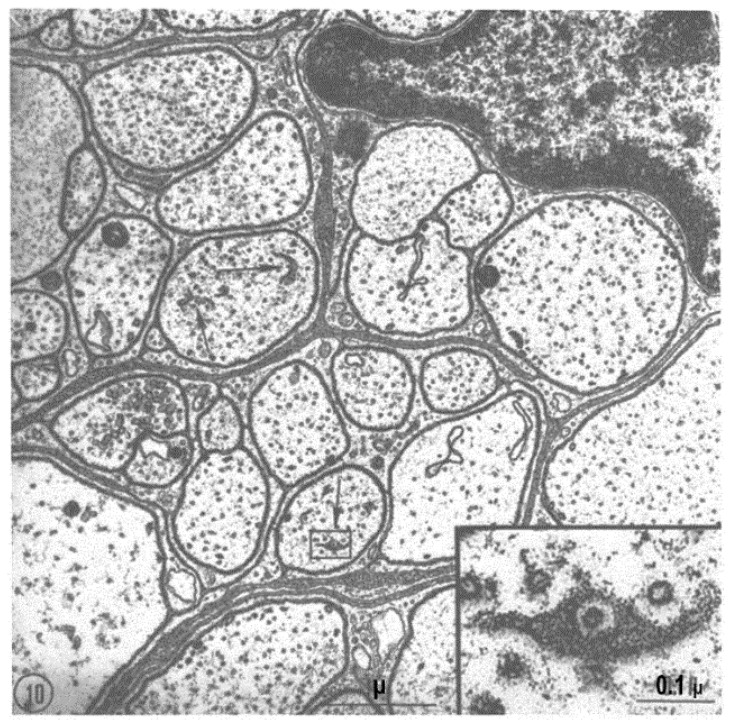
Cross-sectioned small axons (0.2–3.0 μm in size) of a root of the 6th abdominal ganglion. Several axons are enveloped by invaginations of the same satellite cell. The arrows point to small areas of FS, one of which shows a pore (~550 A in size, arrow) containing a microtubule (see inset). Adapted from Ref. [6].

**Figure 11 ijms-24-05385-f011:**
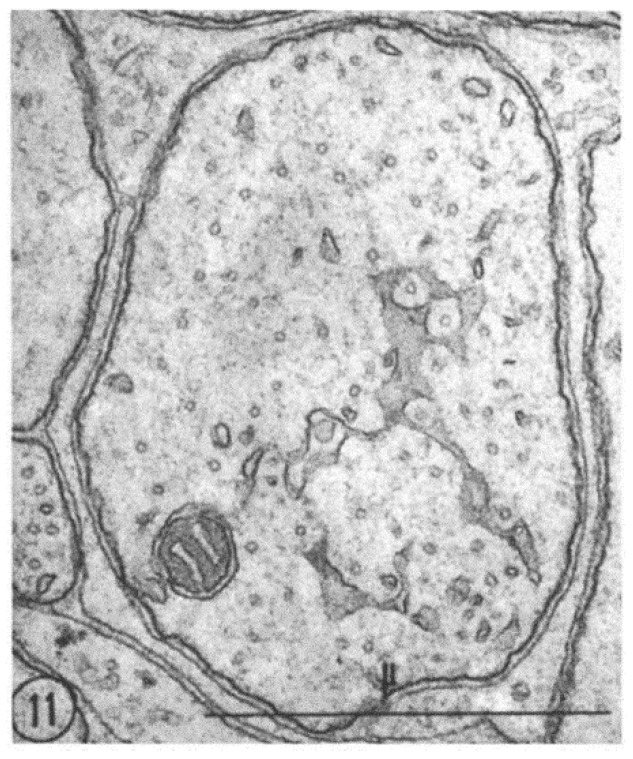
Cross-sectioned small axon displaying a portion of FS with pores containing microtubules. Adapted from Ref. [6].

**Figure 12 ijms-24-05385-f012:**
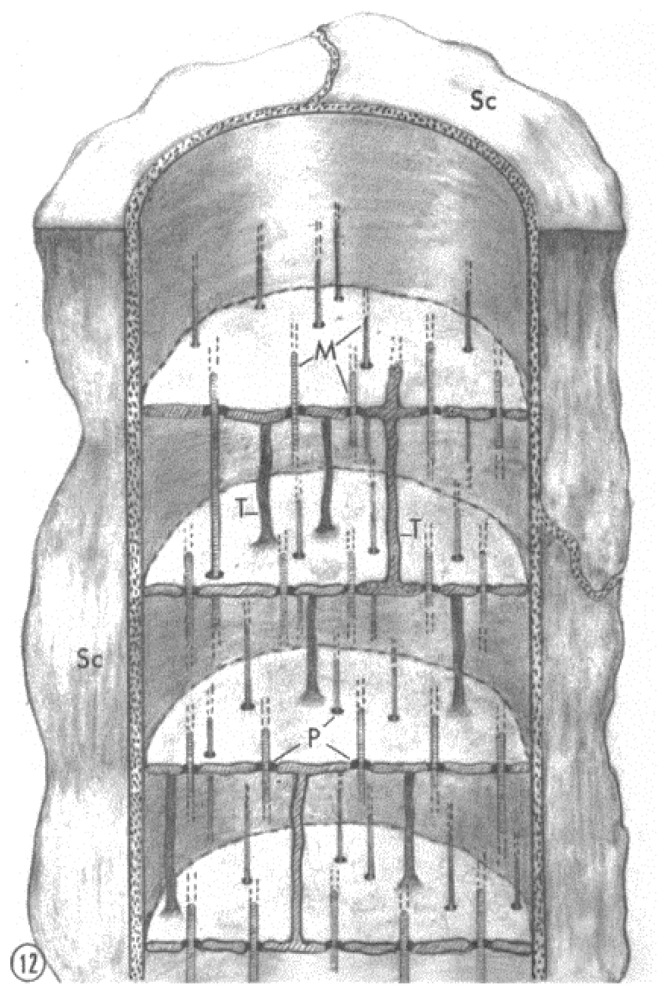
Schematic diagram of FS in a small axon. In this diagram, each septum is shown to be continuous across the axon, but in fact it is likely to be interrupted by large openings to allow the flow of mitochondria and other organelles. Indeed, in larger fibers, wide discontinuities in the septa in the central areas were observed, as discussed in the text. Note that for clarity mitochondria are not shown. T, longitudinal membranous tubules; M, microtubules; P, pores; Sc, satellite cells. Adapted from Ref. [6].

**Figure 13 ijms-24-05385-f013:**
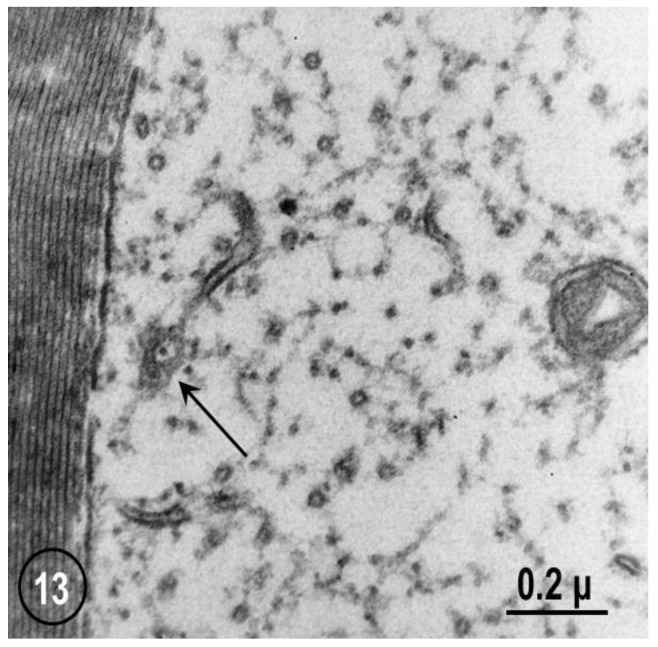
Cross-sectioned myelinated axon of mouse sciatic nerve that displays a small area of a FS, showing a pore containing a neurofilament (arrow). Reported as “C. Peracchia, unpublished observation” in Ref. [6].

**Figure 14 ijms-24-05385-f014:**
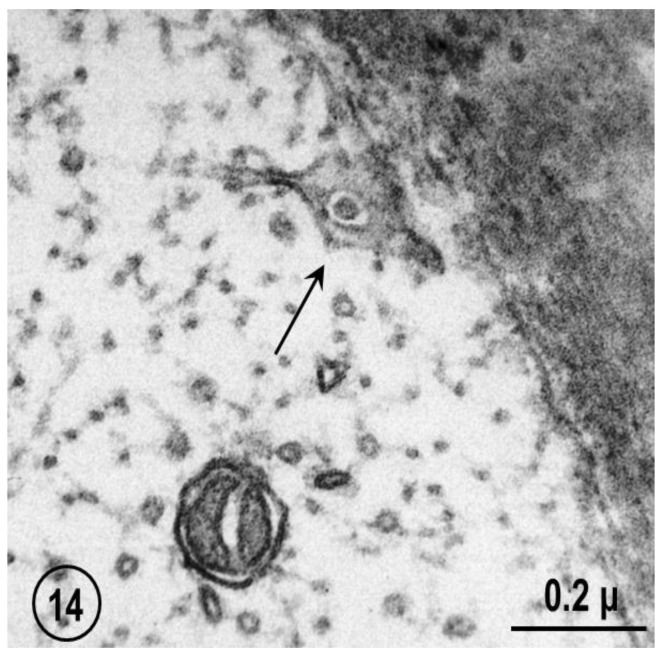
Cross-sectioned myelinated axon of mouse vagus nerve that displays a portion of a FS showing a pore containing a microtubule (arrow). Reported as “C. Peracchia, unpublished observation” in Ref. [6].

**Figure 15 ijms-24-05385-f015:**
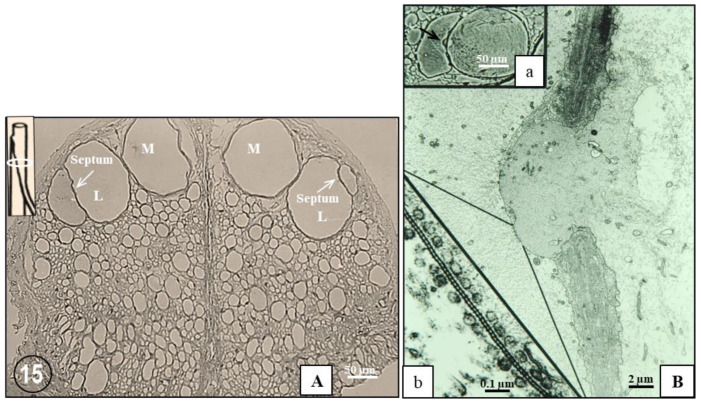
(**A**). Phase contrast image of cross-sectioned crayfish nerve cord showing median (M) and lateral (L) giant axons. The lateral axons are sectioned at the septum, which joins two axonal segments (inset, white circle). (**B**). Electron micrograph showing the caudal segment of a lateral giant axon penetrating through the septum to form gap junctions with the cranial segment; this junctional area is also seen in a phase contrast micrograph (inset (**a**), arrow). At higher magnification (inset (**b**)), the cross-sectioned gap junction shows a beaded profile. The beads are innexin-made junctional hemichannels (innexons) that bind to each other across the extracellular gap. The channels’ center-to-center spacing is ~200 Å. The junction is bordered on both sides by rows of 500–800 Å vesicles.

**Figure 16 ijms-24-05385-f016:**
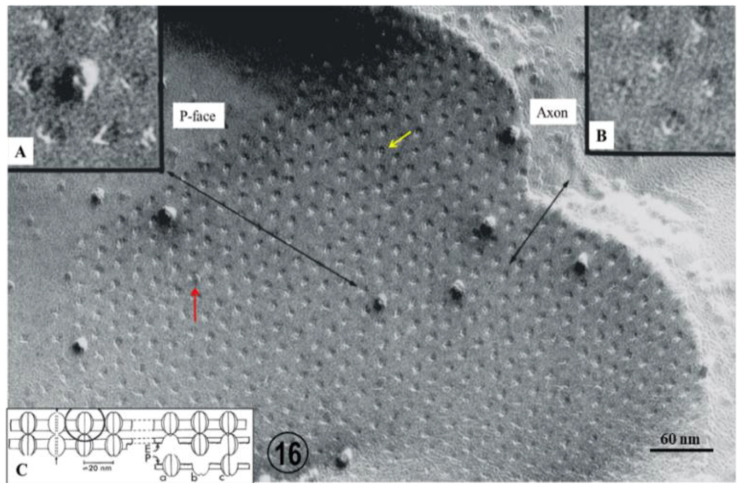
Freeze fracture replica of a crayfish gap junction (P = Protoplasmic face) between two segments of a lateral giant axon. Pits 60–100 Å in size are arranged in an irregular hexagonal array with a unit cell of −200 Å. Some pits display a central bump ~25 Å in size. A few pits are occupied by innexons ~125 Å in size. The innexons contain a central depression ~25 Å in size which represents the opening of a channel’s pore (insets **A** and **C**). Inset (**B**) shows a region (P-face) in which a pentagonal arrangement of pits which contain a small central particle which is the negative image of the pore opening (see the yellow arrow in the main image and black arrow in Inset (**C**). In inset C, E = Exoplasmic face and P = Protoplasmic face, and the circle indicates an innexon. The red arrow points to a monomer of an innexon which is split lengthwise (see “c” in inset (**C**). Adapted from Ref. [33].

**Figure 17 ijms-24-05385-f017:**
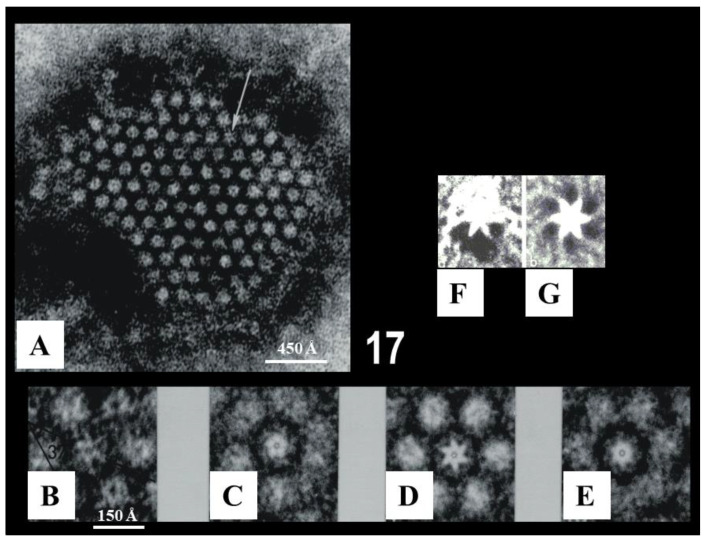
Isolated crayfish junction, negatively stained with phosphotungstic acid (**A**). Innexons, ~95 Å in size, are packed in a hexagonal array with a unit cell of ~155 Å. Many innexons contain a central electron-opaque dot ~20 Å in size representing the hydrophilic pore (opening) of the cell–cell channel. Some innexons (white arrow in (**A**)) display the six radially arranged innexins. The photographic rotation method (see Ref. [34]), applied to the innexon seen in (**A**) (arrow), confirms the six-subunit arrangement (**B**–**E**). Five-step (**C**) and seven-step (**E**) rotations of the innexon produce blurred images, while a six-step rotation (**D**) clearly shows six innexins resulting in a six-pointed star image. Similarly, the six innexins are seen in a freeze-fracture image (**F**) rotated six times (**G**). Adapted from Ref. [33].

**Figure 18 ijms-24-05385-f018:**
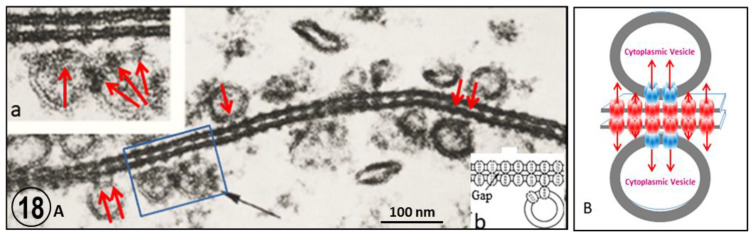
A Thin section of a gap junction between crayfish lateral giant axons. The cross-sectioned junctional membranes (**A**) display a beaded profile due to the presence of innexons. The membranes are coated with 500–800 Å vesicles (black arrow) whose membranes also contain particles (red arrows in (**A**) and inset (**a**)) similar in size to junctional innexons; often these particles appear to join to junctional innexons ((**A**) and insets (**a**,**b**)). Occasionally, neighboring vesicles appear to bind to each other via particle–particle interaction (inset (**a**)). The hypothesis is that the vesicles may establish direct communication with each other and with vesicles lining the other side of the junction (**B**). However, the author is aware that there is no experimental evidence that connexons/innexons can bind to each via their cytoplasmic domains. Adapted from Ref. [35].

**Figure 19 ijms-24-05385-f019:**
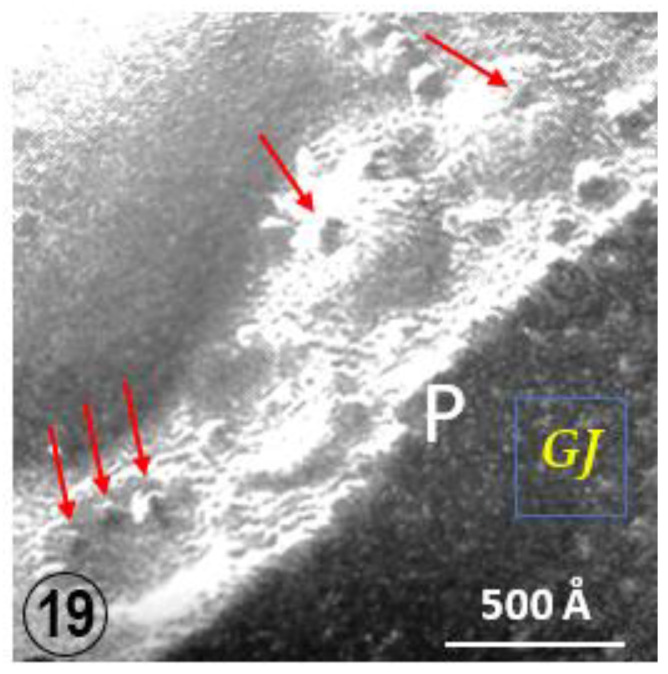
Freeze fracture image of a gap junctions (GJ) between crayfish lateral giant axons. The neighboring vesicles contain particles and pits (red arrows) similar in size to the junctional particle. P = P face, E = E face.

**Figure 20 ijms-24-05385-f020:**
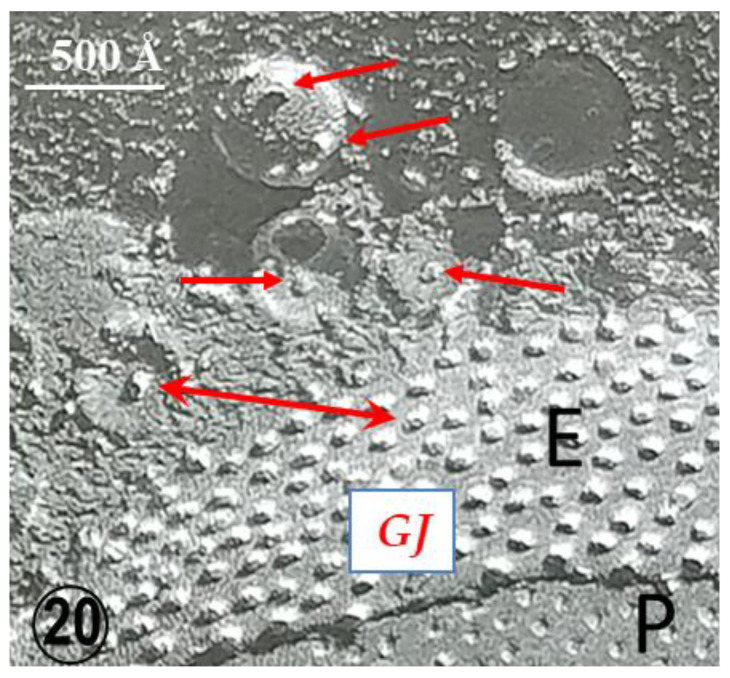
Freeze fracture image of a gap junctions (GJ) between crayfish lateral giant axons. The neighboring vesicles contain particles and pits (red arrows) similar in size to the junctional particle. Often the particles in the vesicles display a central dimple similar in size to that of the junctional membranes (double-headed arrows) P = P face, E = E face.

**Figure 21 ijms-24-05385-f021:**
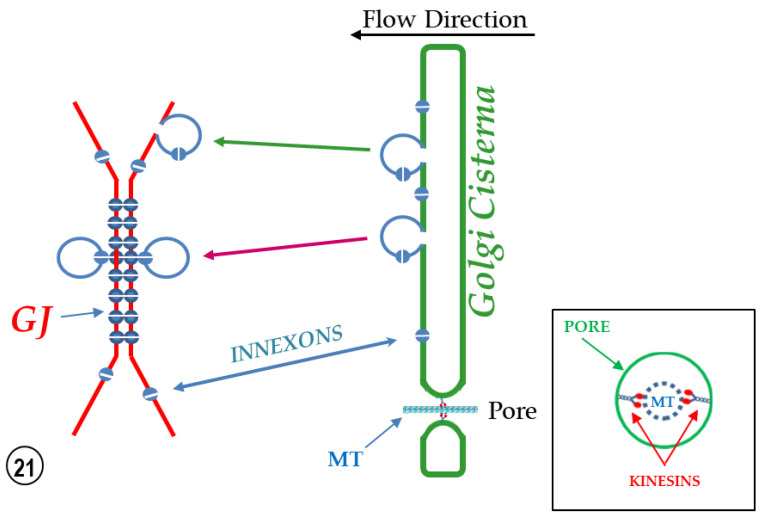
Schematic diagram of potential role of Golgi cisternae in gap junction channel and hemichannel formation in lateral giant axons of crayfish. It is believed that innexon-containing vesicles bud off Golgi cisternae (red and green arrows). Some of the vesicles would fuse with the plasma membrane, inserting in it innexon hemichannels (green and blue arrows). Other vesicles would bind to the gap junction (GJ) via innexon–innexon interaction (red arrow). The lower end of the cisternae shows a cross-sectioned pore containing a microtubule linked to the pore by kinesins. The inset shows our interpretation of the interaction between kinesins and the microtubule (MT) at an FS’s pore.

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
