# Peer review of "Potential Role of Fenestrated Septa in Axonal Transport of Golgi Cisternae and Gap Junction Formation/Function"

_ijms, 2023, doi:10.3390/ijms24065385_

Round 1
Reviewer 1 Report
This is a very nice review of a body of knowledge, first collected in the 1970's (the golden age of electron microscopy), that deserves greater attention and study, in particular through application of modern molecular approaches and in light of what has been learned over the decades about axon structure and function. The knowledge in question is the evidence of fenestrated septa within axons. The anatomical descriptions (using conventional light and electron microscopy, and some freeze-fracture) provide a clear understanding of these structures. The origin (from Golgi) and functions (possibly a source of gap junction protein at axon terminals) is more speculative, but is a foundation on which the original anatomical observations could drive future research. I found only a few minor places where minor corrections are required:
Page 2, line 93-94 "...and that they particpate..."
Page 7, line 179 "...that is like the repeat..."
Page 13, lines 327-329. I wouldn't be so negative! How about: "FS were described in detail in crayfish axons [6], but were also identified in various vertebrate axons as well [REFs?], suggesting a role in nerves in general."
Author Response
Answers to reviewer #1.
The author is grateful for the helpful comments of the reviewer and has modified the paper according to his/her comments.
Reviewer #1
This is a very nice review of a body of knowledge, first collected in the 1970's (the golden age of electron microscopy), that deserves greater attention and study, in particular through application of modern molecular approaches and in light of what has been learned over the decades about axon structure and function. The knowledge in question is the evidence of fenestrated septa within axons. The anatomical descriptions (using conventional light and electron microscopy, and some freeze-fracture) provide a clear understanding of these structures. The origin (from Golgi) and functions (possibly a source of gap junction protein at axon terminals) is more speculative, but is a foundation on which the original anatomical observations could drive future research. I found only a few minor places where minor corrections are required:
Page 2, line 93-94 "...and that they particpate..." Done!
Page 7, line 179 "...that is like the repeat..." Done!
Page 13, lines 327-329. I wouldn't be so negative! How about: "FS were described in detail in crayfish axons [6], but were also identified in various vertebrate axons as well [REFs?], suggesting a role in nerves in general." Done!

Reviewer 2 Report
The hypothesis paper by Peracchia postulates that fenestrated septa are a part of the Golgi apparatus and play an important role in delivering gap junction hemichannels (connexons/innexons) to the plasma membrane in the axons of neurons. The hypothesis is based on light and electron microscopic investigations performed by the Peracchia lab many years ago, however the hypothesis -as the author states- never received significant attention or was experimentally approached. Thus, the nature of fenestrated septa and whether they play a role in the trafficking of gap junction proteins remains unclear. The author revisits the topic to re-draw attention to the issue in hope to stimulate interest and future research on the topic. To revisit the topic appears appropriate, as these septa also are present in vertebrate axons and deserve further characterization. The EM images are quite amazing and are a reminder that this technique can still provide valuable information, e.g. if combined with antibodies, life-cell imaging (CLEM, correlative light-electron microscopy), and AI-based protein structure predictions.
Major:
The concept of “triple gap junctions” (Figure 18B, 21, and p. 12) appears overly ambiguous. The formation of triple gap junctions requires that innexons interact and dock via their intracellular domains. Is their published evidence – besides the single EM image presented in Figure 18A- suggesting such an interaction? To my knowledge, recent cryo EM particle analyses do not show docking of connexons or channels via their intracellular C-terminal domains. More importantly, what would be the benefit of forming triple GJs? What can they do that normal GJs cannot do? Bringing cytoplasmic content from one cell to the other can be achieved by regular GJ channels. Why do they need to be extended? The concept, as it currently is described appears rather distractive, and if kept (it is not related to the main question and thus not needed) should be substantiated and better explained.
Throughout: As there is only one author, “we” should be changed into “I”. I understand that the study has not been done by a sole author, however as the article is not co-authored by these co-workers, it solely reflects the view of the author.
Fig. 9: Top and bottom axons are hard to be recognized by the unexperienced reader. Either outline axons, or better describe such as “Top and bottom axons separated by elongated mitochondria appearing electron dense”.
L. 299: microtubule (MT), not mitochondrion
L. 316: add “in the ER” to read “… in the ER and Golgi” as not all Cxs oligomerize in the Golgi (see H. Evans, P. Martin, M. Falk)
L. 326: I suggest changing “has been largely ignored” to “has not been investigated”, or similar
Minor:
L. 23: Semicolon between ‘innexins’ and ‘connexins’
L. 32: ‘we propose’ instead of proposes
L. 42: ‘Kinesin’, not kinesins
L. 80ff: species name not capital but lower case (oceanicus, americans, domestica)
L. 84: Panaceus japonicum, cursive and japonicum in lower case
L. 94: add ‘they’ to read ‘… that they participate …’
L. 179: ‘like’, not loke; better would be ‘similar to’
L. 220: ‘nerve’, not verve
L. 297: ‘cisternae’, not cistern
L. 308: ‘vesicle to vesicle’, not vesicles
L. 341: add “to” to read “… be used to determine …”
L. 345: The author declares no … (there is only one)
L. 347ff: The title of some references is printed in cursive
Author Response
Answers to reviewer #2.
The author is grateful for the helpful comments of the reviewer and has modified the paper as closely as possible to his/her comments.
Reviewer #2
The hypothesis paper by Peracchia postulates that fenestrated septa are a part of the Golgi apparatus and play an important role in delivering gap junction hemichannels (connexons/innexons) to the plasma membrane in the axons of neurons. The hypothesis is based on light and electron microscopic investigations performed by the Peracchia lab many years ago, however the hypothesis -as the author states- never received significant attention or was experimentally approached. Thus, the nature of fenestrated septa and whether they play a role in the trafficking of gap junction proteins remains unclear. The author revisits the topic to re-draw attention to the issue in hope to stimulate interest and future research on the topic. To revisit the topic appears appropriate, as these septa also are present in vertebrate axons and deserve further characterization. The EM images are quite amazing “ thanks” and are a reminder that this technique can still provide valuable information, e.g. if combined with antibodies, life-cell imaging (CLEM, correlative light-electron microscopy), and AI-based protein structure predictions.
Major:
The concept of “triple gap junctions” (Figure 18B, 21, and p. 12) appears overly ambiguous. The formation of triple gap junctions requires that innexons interact and dock via their intracellular domains. Is their published evidence – besides the single EM image presented in Figure 18A- suggesting such an interaction? To my knowledge, recent cryo EM particle analyses do not show docking of connexons or channels via their intracellular C-terminal domains. More importantly, what would be the benefit of forming triple GJs? What can they do that normal GJs cannot do? Bringing cytoplasmic content from one cell to the other can be achieved by regular GJ channels. Why do they need to be extended? The concept, as it currently is described appears rather distractive, and if kept (it is not related to the main question and thus not needed) should be substantiated and better explained.
I agree entirely with the reviewer that the idea of a “triple gap junctions” is very speculative, especially since there is no experimental evidence that connexons/innexons can bind to each other via cytoplasmic domains. Therefore I have eliminated the words “triple gap junctions” and mentioned in several places that the idea is very speculative and needs to be tested experimentally.
For example, in the legend of Fig. l8 I added:
“My hypothesis is that the vesicles may establish direct communication with each other and with vesicles lining the other side of the junction (B). However, I am aware that there is no experimental evidence that connexons/innexons can bind to each via their cytoplasmic domains.
Also, in line 293 on I added the following:
I do understand that this hypothesis is very speculative because there is no experimental evidence that connexon/innexons can interact by their cytoplasmic domains. In 2020 I suggested possible interaction between domains of the cytoplasmic loop of Cx32, Cx43 and innexin-1 [36], but once again these potential interactions are entirely speculative.
Throughout: As there is only one author, “we” should be changed into “I”. I understand that the study has not been done by a sole author, however as the article is not co-authored by these co-workers, it solely reflects the view of the author. Done!
Fig. 9: Top and bottom axons are hard to be recognized by the unexperienced reader. Either outline axons, or better describe such as “Top and bottom axons separated by elongated mitochondria appearing electron dense”. Done!
- 299: microtubule (MT), not mitochondrion Done!
- 316: add “in the ER” to read “… in the ER and Golgi” as not all Cxs oligomerize in the Golgi (see H. Evans, P. Martin, M. Falk) Done!
- 326: I suggest changing “has been largely ignored” to “has not been investigated”, or similar Done!
Minor:
- 23: Semicolon between ‘innexins’ and ‘connexins’
- 32: ‘we propose’ instead of proposes
- 42: ‘Kinesin’, not kinesins
- 80ff: species name not capital but lower case (oceanicus, americans, domestica)
- 84: Panaceus japonicum, cursive and japonicum in lower case
- 94: add ‘they’ to read ‘… that they participate …’
- 179: ‘like’, not loke; better would be ‘similar to’
- 220: ‘nerve’, not verve
- 297: ‘cisternae’, not cistern
- 308: ‘vesicle to vesicle’, not vesicles
- 341: add “to” to read “… be used to determine …”
- 345: The author declares no … (there is only one)
- 347ff: The title of some references is printed in cursive.
All done!

Round 2
Reviewer 2 Report
The author has responded appropriately to the critique.